# Fabrication of UV Laser-Induced Porous Graphene Patterns with Nanospheres and Their Optical and Electrical Characteristics

**DOI:** 10.3390/ma13183930

**Published:** 2020-09-05

**Authors:** Jun-Uk Lee, Yong-Won Ma, Sung-Yeob Jeong, Bo-Sung Shin

**Affiliations:** 1Department of Cogno Mechatronics Engineering, Pusan National University, Busan 46241, Korea; lju3534@naver.com; 2Interdisciplinary Department for Advanced Innovative Manufacturing Engineering, Pusan National University, Busan 46241, Korea; decentsoul@naver.com (Y.-W.M.); ysjsykj8025@naver.com (S.-Y.J.); 3Department of Optics and Mechatronics Engineering, Pusan National University, Busan 46241, Korea

**Keywords:** laser-induced plasma (LIP), laser direct writing (LDW), polyimide (PI), structural color, nanosphere, 355 nm pulsed laser

## Abstract

Many studies have been conducted to fabricate unique structures on flexible substrates and to apply such structures to a variety of fields. However, it is difficult to produce unique structures such as multilayer, nanospheres and porous patterns on a flexible substrate. We present a facile method of nanospheres based on laser-induced porous graphene (LIPG), by using laser-induced plasma (LIP). We fabricated these patterns from commercial polyimide (PI) film, with a 355 nm pulsed laser. For a simple one-step process, we used laser direct writing (LDW), under ambient conditions. We irradiated the PI film at a defocused plane −4 mm away from the focal plane, for high pulse overlap rate. The effect of the laser scanning speed was investigated by FE-SEM, to observe morphological characterization. Moreover, we confirmed the pattern characteristics by optical microscope, Raman spectroscopy and electrical experiments. The results suggested that we could modulate the conductivity and structural color by controlling the laser scanning speed. In this work, when the speed of the laser is 20 mm/s and the fluence is 5.28 mJ/cm^2^, the structural color is most outstanding. Furthermore, we applied these unique characteristics to various colorful patterns by controlling focal plane.

## 1. Introduction

Graphene has a single layer, in which a thin layer of carbon has important properties, including a high optical transmittance and electrical conductivity, which make it favorable for use in optoelectronic applications [1,2,3,4]. In addition, graphene has excellent physical properties: mechanical strength and thermal conductivity [5]. Chemical vapor deposition (CVD) and chemical exfoliation are commonly used in graphene manufacturing methods. However, as the fabrication of graphene patterns is very expensive, many researchers have attempted to develop an inexpensive process for applying it to circuits [6]. The so-called laser-induced graphene (LIG) and laser-induced porous graphene (LIPG) on polyimide (PI) film, of which the 2D peak is centered at 2700 cm^−1^, like single-layer graphene, is an alternative, because it can be produced quickly and inexpensively, and it is also easy to fabricate patterns with it [7,8,9,10,11]. The interaction of PI and the laser is broadly classified as a delamination effect or a carbonization effect, and it is closely related to the laser fluence [12,13,14]. The delamination effect (photochemical effect) occurs when the UV laser power is weak, while the carbonization effect (photothermal effect) occurs as the laser fluence increases. As the amount of energy increases, part of the PI is evaporated by physical explosive and chemical decomposition. Photothermal phenomenon occurs due to the high thermal energy caused by the laser. In the case of carbonization by a 355 nm UV pulsed laser, photothermal and photochemical effects occur simultaneously, but the main effect is the photothermal effect. The main role of the photochemical effect by UV region laser is dissociation of the C–N bond on the PI surface, of which the UV laser allows higher carbonization to be achieved, even at low temperatures. However, many studies have been conducted on LIG, using the carbonization effect, with lasers of various wavelengths, because high thermal energy can also break the C–N bond. It is common for the LIG to be black because it has a multilayer structure and high absorption of visible light. However, the commonly used single- or few-layer graphene is transparent due to the Pauli barrier effect [15,16]. Specific colors have often been reported in multilayer graphene researches [17,18]. In general, the structure color is caused by film interference (reflection of light waves by the upper and lower boundaries), scattering (small particles; Mie scattering and Rayleigh scattering), diffraction grating(periodic structure) and photonic crystals(periodic structure) produced by microstructures or nanostructures [17,19,20,21]. However, the colors of graphene and graphene-oxide multilayers on various dielectric layers have been explained through film interference, and they have been mainly studied at tens of nm thickness. The effects of the material thickness and the types of dielectric layers have also been analyzed. The reflectance varies according to the thickness or the dielectric layer, but the reflectance in the blue region near 400 nm is high.

In this paper, we present, for the first time, graphene flakes and graphitization patterns based on LIPG with nanospheres which showed structural colors based on reflectance of the graphene multilayers and the high conductivity of the 3D carbon network. Our main novelty is the observation of structural color (light scattering from the porous and nanostructured films after UV laser irradiation). Our LIPG patterns, which have the chemical composition of the existing results of conductivity that were maintained, were fabricated through laser speed control based on a high overlap rate, which created laser-induced plasma (LIP). The surface properties of the LIPG patterns were confirmed by Raman, and the structural shape was observed by FE-SEM. A comparison was made according to the laser scanning speed, to identify the optimum conditions of conductivity and structural color. The colorization of LIPG with nanospheres shows another approach, and the discussion of LIPG colors that are often seen begin here.

## 2. Materials and Methods

### 2.1. Laser System

In this experiment, we used a laser scribing system equipped with a 355 nm pulsed laser (AONano 355-5-30-V from Advanced Optowave, Ronkonkoma, NA, USA) as an experimental tool. The specifications of the laser are as shown in Table 1, and a schematic illustration of the laser system configuration is shown in Figure 1a. The pulse of the laser expanded through the beam expander, then moved through the movements of the mirrors of the galvano scanner (hurrySCAN III 14 from SCANLAB, Pucheim, Germany) and finally passed through the F-θ lens, f: 105.9 mm (S4LFT4100/075 Telecentric Scan Lens from Sill Optics, Wendelstein, Germany). With this system, we used the laser direct writing (LDW) method. LDW uses laser beam to create a controlled localized damage on the surface [22]. We designed the patterns by using a 2D CAD (AutoCAD, San Rafael, CA, USA), which has an advantage in that various patterns can be implemented in one step. Figure 1b is an actual photograph of when a 355 nm pulsed laser irradiates PI. We observed that the LIP phenomenon was caused by the intense heat and pressure of the laser. LIP is the plasma produced by the interaction of high-energy laser pulses with matter in any state of aggregation [23]. The LIP increased when the speed of the laser was slow, because of the high pulse-overlap rate.

### 2.2. Principle of Photochemical Ablation of PI, Using a 355 nm Pulsed Laser

PI is basically a polymer containing an imide group. The PI contains a strong charge transfer complex (CTC). The charge transfer complex is a kind of intermolecular force and is an attractive force between an electron donor and an electron acceptor [24]. Figure 2a shows the polyimide chemical structure, which is very stable as electrons are pushed from diamines with electron donors to carbonyl groups with electron acceptors. This phenomenon occurs not only among adjacent units in the chain but also between polymer chains, so the polyimide has a layered structure. Due to the strong bonding of these charge transfer complexes, polyimide has high strength, but most of the polyimides have the disadvantages of being insoluble and infusible, so they have poor moldability and processability; on the other hand, polyimide has a very high absorption rate with respect to UV light [24]. Although the absorption rate varies depending on the fluence of the laser, it has an absorption rate of over 85% from very low fluence of 10 mJ/cm^2^ [25]. There have been many studies on the processing of polyimide, using UV lasers [26,27].

The wavelength of the laser used in the experiment was 355 nm, which has a photon energy of 3.5 eV. This energy can only break C–N bond bonds among PI’s chemical bonds, as shown in Table 2. Figure 2b shows the photochemical ablation that occurs with laser irradiation of a 355 nm pulsed laser. Photochemical ablation is due to laser photons detaching atoms or groups directly [28]. However, if the photothermal effect increases the surface’s temperature with UV laser irradiation, polymer bond energy is lower than steady state, and PI is vaporized as LIP, as shown in Figure 2c [29]. LIP is plasma ablation.

Since we used LDW by carrying a laser beam, the beam does not stay in one place and proceeds. We observed that processing at focus resulted in the ablation of the LIPG [30]. To counter that, we intentionally defocused the beam, to increase the size of the laser beam, producing lower power densities. As the beam size increased, the overlap rate of the pulses increased. Figure 3a gives a schematic illustration of a defocused beam (−4 mm away from the focused beam). Commercial PI film (DUPONT™ KAPTON^®^ HN) with a nominal thickness of 125 μm was irradiated at a defocused plane (−4 mm). Figure 3b gives a schematic illustration of the overlapping of a 355 nm pulsed laser. The blue line shows the laser path. In this case, a unidirectional scanning strategy was used for a one-step processing. Through this, fabrication was possible at a faster speed. The spacing of the laser scribing line pattern was 50 μm. Because of the short distance of the line patterns, the pulse-overlap rate increased. We expect that this high pulse-overlap rate made the pattern’s structure unique, which is different from the findings in other studies [30,31,32].

Figure 4 shows the FE-SEM images (JSM-7900F from JEOL, Peabody, MA, USA) of pulse-overlapped line patterns fabricated according to laser speed, by using a defocused beam. As shown in Figure 4a–e, the slower the laser speed, the larger the spot size and the higher the fluence (Table 3 shows the laser speed and fluence according to the speed). As shown in Figure 4f–j, we observed the delamination effect according to the laser speed. Figure 4k–o shows that, smaller the laser speed, the rougher the surface. From Figure 4, it can be seen that delamination effect (photochemical effect) and carbonization effect (photothermal effect) can be adjusted by adjusting the laser speed, to adjust the fluence [12,13,14]. Since the line pattern is produced in a short time, thermal energy is small, and there is no pulse overlap in the transverse direction; therefore, we created patterns to increase the thermal effects and transverse pulse overlap.

## 3. Results and Discussion

### 3.1. Morphological Characterization

We fabricated 1 × 1 cm^2^ square patterns, using the same experimental conditions (line spacing: 50 μm, overlapping of laser pulses and defocused beam) as described above. The square patterns were produced by adjusting the speed. Speeds were set to 20, 40, 60, 80 and 100 mm/s. Figure 4 shows the FE-SEM images of pulse-overlapped patterns on PI. Figure 4a,f,k shows patterns fabricated when the laser speed was 100 mm/s. Figure 4f,k confirms that the polyimide peels off. Our analysis revealed that the layered structure of the polyimide by the CTC was related to the polyimide-peeling phenomenon. Figure 4b,g,l shows the patterns fabricated when the speed was 80 mm/s. Figure 4g,l shows that there was more of a delamination effect. It is clear that the high overlapping rate increased the delamination effect. Figure 4c,h,m shows the patterns fabricated when the speed was 60 mm/s. At this laser speed of 60 mm/s, the patterns became porous. We predicted that, when the speed was 60 mm/s, the peeling surface layer would be removed by the high fluence, and the photothermal-ablation phenomenon would be increased due to the strong LIP phenomenon. Figure 4d,i,n shows the patterns fabricated when the speed was 40 mm/s. As shown in Figure 4i, more pores were formed, and it seems that the peeled surface layers were almost removed. This is because the LIP increased further as the speed slowed down. Figure 4n shows that spherical structures were formed on the rough surface.

Figure 5e,j,n shows the patterns fabricated when the speed was 20 mm/s. Due to the strong fluence, most of the peeled surface layers disappeared. This means that the thermal effect increased, rather than the chemical effect. The dominant photothermal effect caused carbonaceous materials to aggregate, which produced sphere structures. It was expected that the LIP induced by the strong thermal effect (gaseous carbon plasma [33]) was agglomerated as a result of the rapid temperature drop on the surface. As shown in Figure 5o, we observed nanospheres on the pattern’s clean surface.

The FE-SEM images in Figure 5 confirmed that pores formed on the surface, from a laser speed of 60 mm/s. We took FE-SEM transection images to observe the pores in more detail. Figure 6’s FE-SEM transection images of the patterns confirm that a 3D carbon network formed. Figure 6a–c shows the patterns fabricated when the laser speed was 60, 40 and 20 mm/s. The slower the speed, the thicker the 3D carbon network. This occurred because, as the LIP phenomenon increased, strong fluence and pressure were applied to the PI. Strong heat and pressure cause non-carbon atoms, namely H, N and O, to escape from the PI, to form H_2_, H_2_O, N_2_ and NH_3_. Carbon atoms made up the graphene layer [34]. The thickness of the 3D carbon network increased from the laser speed of 60 mm/s, but the thickness of the total thickness of patterns decreased (as shown in Table 4). It is because of photothermal ablation based on heat-induced melting [35].

### 3.2. Electrical and Optical Characterizations

Figure 7a–d shows optical micrograph images (BX60M from OLYMPUS, Shinjuku, Japan) of the patterns. The light reflection phenomenon increases when the speed decreases. When the speed is 60 mm/s, the reflection phenomenon increases rapidly. We think it is because of the porous 3D carbon network fabricated from a laser speed of 60 mm/s. Figure 7e is a graph of the conductivity of patterns created according to laser speed. As the scanning speed of the laser decreases, the conductivity tends to increase. This again shows that, as the scanning speed of the laser decreases, the heat and pressure in the PI increase. It means that high fluence produces graphitization more effectively. However, when the laser speed was 20 mm/s, there was a slightly lower conductivity than when it was 40 mm/s. It means that, when the laser’s speed is 20 mm/s, the energy is so high that the 3D carbon network is broken and blown away. Therefore, we have confirmed that the speed of 40 mm/s is the speed of the highest conductivity value. However, this strong energy at a laser speed of 20 mm/s creates new shapes on the surface. As shown in Figure 7f, we observed the unique colorful patterns. We predicted that it is because of the structural color of LIPG patterns fabricated by strong heat and pressure.

Figure 8a demonstrates the I-V characteristics of patterns with a voltage range from −10 to 10 V. These electrical characteristics were measured by the source meter (KEITHLEY, Source Meter 2450, Tektronix Co., Solon, OH, USA). The graph shows different gradients according to the laser speed. However, all the patterns show stable electrical properties. Through this, it can be predicted that a conductive carbon network is well formed. However, the gradient at laser speed of 40 mm/s is more inclined than at laser speed of 20 mm/s. This is due to explosive gas release by strong pressure and fluence at the laser speed of 20 mm/s. We expect that the explosive gas release based on photothermal effect causes the 3D carbon network to be broken. Figure 8b shows the Raman spectra analysis (NRS-5100, Easton, MD, USA) with laser speed. All of the resulting samples, according to laser speed, were characterized by Raman spectroscopy. It is clear that three peaks of all the samples stood out (a D peak near 1350 cm^−1^, a G peak near 1580 cm^−1^ and a 2D peak near 2700 cm^−1^ [36,37]). In general, the ratio $I_{D}/I_{G}$ is usually used to substantiate the degree of graphitization of carbon-based materials (the lower the ratio is, the greater the crystallinity) [38,39]. Therefore, the $I_{D}/I_{G}$ values (Table 5) were also used to specify the graphitization level. They are laser-speed-dependent almost linearly. The patterns fabricated at 20 mm/s ($I_{D}/I_{G}\;$= 0.94) has the highest content of graphite like the structure of $sp^{2}$ microdomains and nanographite crystals, when compared to the patterns fabricated at 100 mm/s ($I_{D}/I_{G}\;$= 1.42) [40]. Therefore, we deduce that $sp^{3}\;$-carbon atoms in PI are photo-thermally converted to $sp^{2}\;$-carbon atoms as the laser speed decreases [37]. We think that nanographite crystals make patterns to photonic structure.

### 3.3. Application Structural Color Based on Nanospheres Fabricated by Controlling Focal Plane

We arranged the focal plane to fabricate various colorful patterns. Except for the focal plane, the conditions are the same as those mentioned in Figure 3. As shown in Figure 9a–j, the pattern was produced by adjusting the focal plane at intervals of 1 mm from +1 to −8. Among images, the pattern produced on the defocused plane (Figure 8g) of −5 showed the most outstanding blue color. Figure 9k shows an actual photo of the structural blue color of patterns fabricated at defocused plane (−5 mm). Figure 9l was taken by FE-SEM image in which we could observe many nanospheres on the surface of the patterns. Figure 9m shows an image of a nanosphere. We predict that multilayer of LIPG with nanospheres induce interference based on multiple scattering [41]. Because our blue iridescence is similar to the structural coloration in nature: the blue butterfly *Morpho rhetenor* [42]. Therefore, using our laser processing method, it was possible to easily produce structural colors based on LIPG.

## 4. Conclusions

We used a 355 nm pulsed laser to fabricate graphitization patterns with nanospheres that make patterns with distinctive properties, such as conductivity, flexibility and structural color. Our patterns with nanospheres were fabricated by irradiating them onto the commercial PI film. The laser beam was defocused by −4 mm, from the focal plane, which caused the LIP phenomenon in the PI, and patterns were fabricated at laser scanning speeds of 100 to 20 mm/s, at 20 mm/s intervals. We analyzed the morphological characteristics of pulse-overlapped patterns according to speed. FE-SEM observation confirmed that nanospheres were formed on the PI, at a laser condition from the laser scanning speed of 40 mm/s, and 3D carbon networks of LIPG were produced from the laser scanning speed of 60 mm/s. We analyzed the pattern’s characterizations through electrical measurements and Raman spectroscopy. Through the results of this experiment, we showed the possibility that electrical and optical properties can be changed through 355 nm pulsed laser’s condition, and we presented a new approach of laser-induced structural color with laser processing.

## Figures and Tables

**Figure 1 materials-13-03930-f001:**
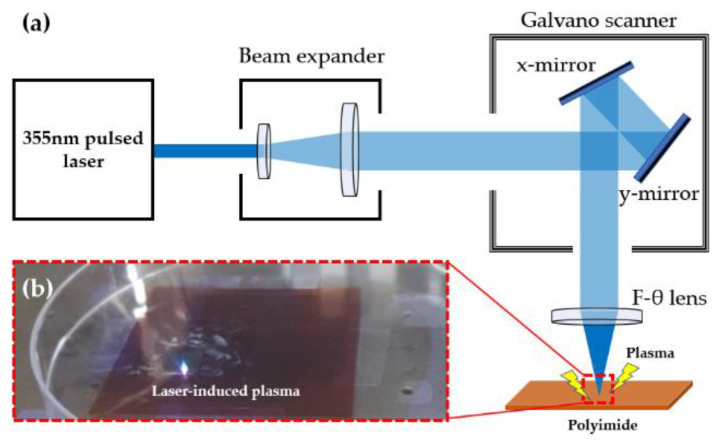
(**a**) Schematic illustration of the laser system. (**b**) Photograph of laser-induced plasma on polyimide (PI).

**Figure 2 materials-13-03930-f002:**
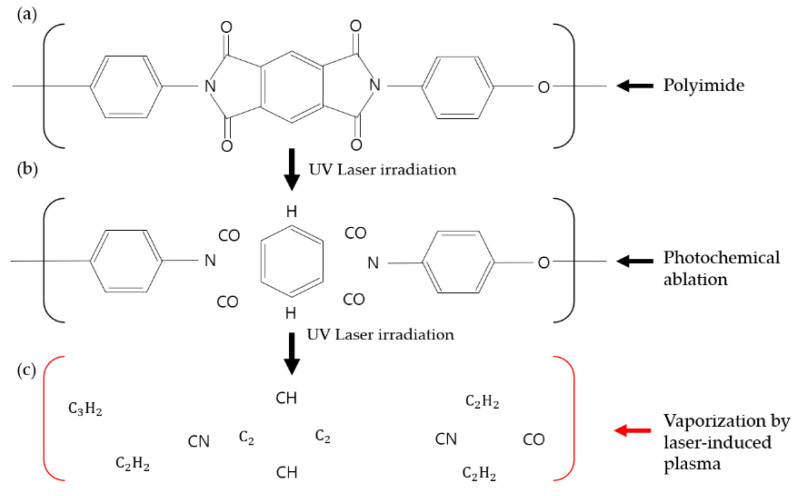
Principle of laser-induced plasma (LIP), using a 355 nm pulsed laser: (**a**) polyimide structure, (**b**) structural change of polyimide after laser irradiation and (**c**) vaporization by plasma generation.

**Figure 3 materials-13-03930-f003:**
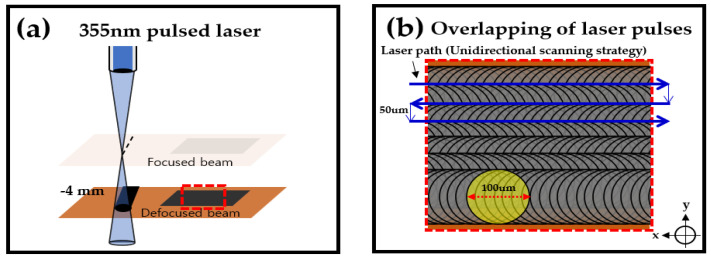
(**a**) Schematic illustration of 355 nm pulsed laser conditions (defocused beam). (**b**) Schematic illustration of the pulse overlap.

**Figure 4 materials-13-03930-f004:**
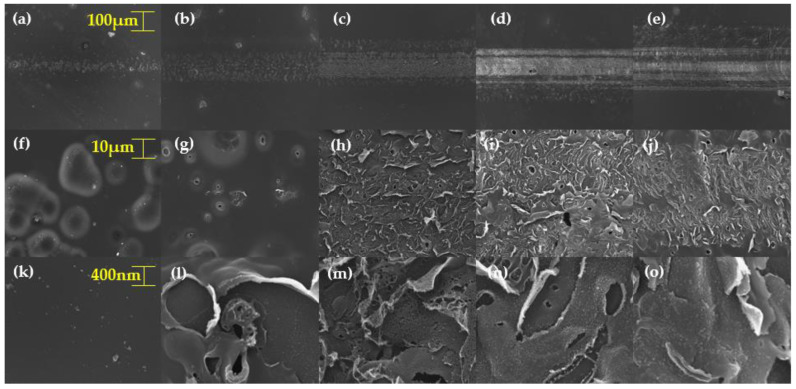
FE-SEM images of pulse-overlapped line patterns fabricated: (**a**,**f**,**k**) at 100 mm/s, (**b**,**g**,**l**) at 80 mm/s, (**c**,**h**,**m**) at 60 mm/s, (**d**,**i**,**n**) at 40 mm/s and (**e**,**j**,**o**) at 20 mm/s.

**Figure 5 materials-13-03930-f005:**
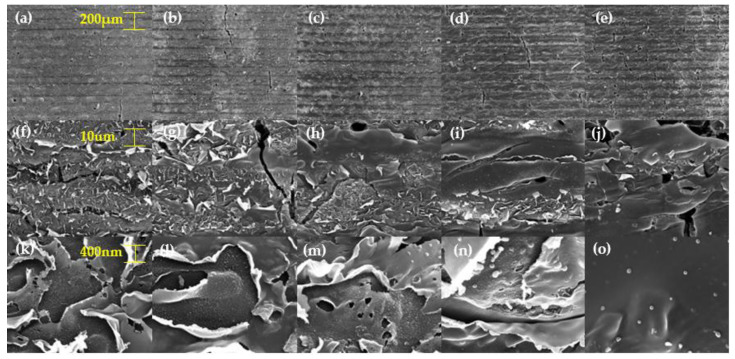
FE-SEM images of pulse-overlapped patterns fabricated (**a**,**f**,**k**) at 100 mm/s, (**b**,**g**,**l**) at 80 mm/s, (**c**,**h**,**m**) at 60 mm/s, (**d**,**i**,**n**) at 40 mm/s and (**e**,**j**,**o**) at 20 mm/s.

**Figure 6 materials-13-03930-f006:**
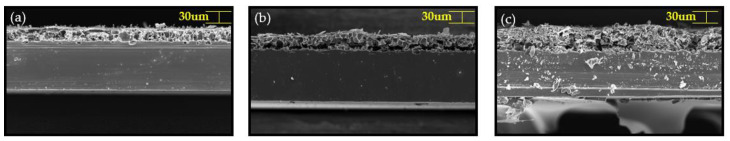
FE-SEM transection images of patterns: (**a**) patterns fabricated at 60 mm/s, (**b**) patterns fabricated at 40 mm/s and (**c**) patterns fabricated at 20 mm/s.

**Figure 7 materials-13-03930-f007:**
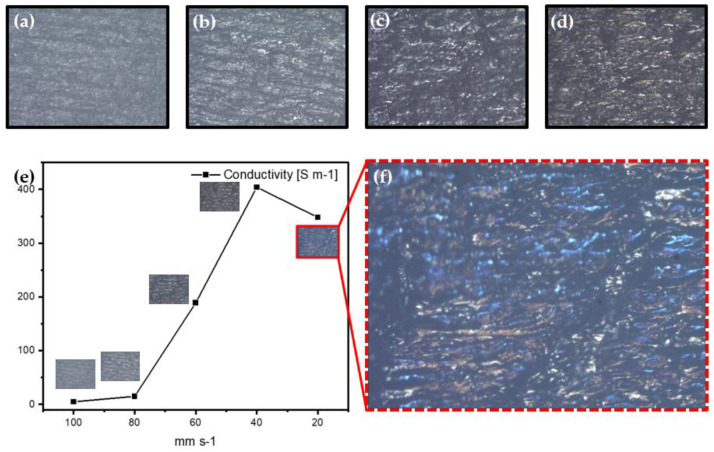
(**a**–**d**) Optical microscope images of overlapped patterns fabricated at 100, 80, 60 and 40 mm/s; (**e**) graph of the conductivity measurement according to laser speed; and (**f**) laser-induced colorful patterns fabricated at 20 mm/s.

**Figure 8 materials-13-03930-f008:**
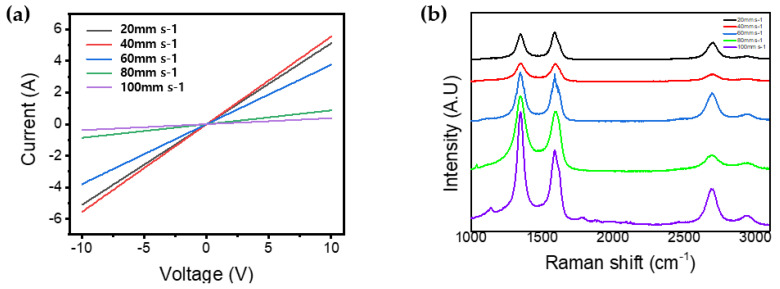
(**a**) I–V characteristics. (**b**) Raman spectra of patterns.

**Figure 9 materials-13-03930-f009:**
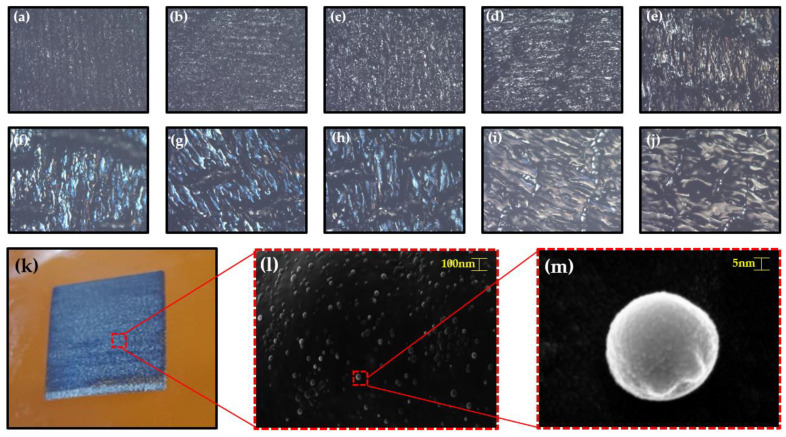
(**a**–**j**) Optical microscope images of patterns fabricated at intervals of 1, from +3 to −7, (**k**) an actual photo of the structural blue color of patterns fabricated at a laser speed of 20 mm/s, (**l**) an FE-SEM image of the patterns, and (**m**) an FE-SEM image of a nanosphere with scattering.

**Table 1 materials-13-03930-t001:** Specifications of the 355 nm UV pulsed laser.

Parameter	Unit	Value
Wavelength	nm	355
Average power	Watt	~2.5
Pulse duration	ns	25
Repetition rate	kHz	30
Mode	-	TEM_00_
Beam diameter	mm	1.5
Beam divergence	mard	<0.5

**Table 2 materials-13-03930-t002:** Chemical bond energies for polymers.

Polymer Bond	C–N	C–H	C≡C	O–O	C=C	C–C	N–N	H–H
Bond Energy (eV)	3.04	4.30	8.44	5.12	6.40	3.62	9.76	4.48

**Table 3 materials-13-03930-t003:** Defocused laser beam condition.

Laser Speed (mm/s)	Spot Size (μm)	Laser Fluence (mJ/cm^2^)
100	35	3.77
80	70	4.15
60	100	4.53
40	120	4.90
20	140	5.28

**Table 4 materials-13-03930-t004:** Thickness of overlapped-pulse patterns.

Laser Speed (mm/s)	Thickness of 3D Carbon Network (µm)	Total Thickness of Pattern (µm)
60	30	130
40	45	125
20	65	120

**Table 5 materials-13-03930-t005:** $I_{D}/I_{G}$ values of the as-prepared patterns calculated from the Raman analysis.

Laser Speed (mm/s)	$I_{D}/I_{G}$
100	1.42
80	1.21
60	1.05
40	1.03
20	0.94

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
