# Peer review of "Fabrication of UV Laser-Induced Porous Graphene Patterns with Nanospheres and Their Optical and Electrical Characteristics"

_materials, 2020, doi:10.3390/ma13183930_

Round 1
Reviewer 1 Report
This manuscript is interesting enough to be published in Materials, however still some remarks need to answered prior to publication and those are listed below:
1. In the introduction section the main motivation, why described LIP approach was used is missing. In this case also where the samples prepared with similar approach can be utilized?
2. Resistance in ohm presented here is not suitable for investigation and discussion. Conductivity measurements need to be performed to be able to provide comparative method to Raman spectroscopy. Unit of Surface Resistance is not comparable to majority of the articles, how the unit ohm/sq is defined? Moreover, difference in resistance is just two orders of magnitude, is it reduction that was obtained similar or comparable also for the rest of the laser-induced approaches? References are missing here.
3. Why the result at 40 mm/s has the best conductivity? Is there any physical reason? Please provide detailed discussion.
4. Please change mm/s to mm s-1. This need to be unified in whole manuscript.
5. Authors claimed that they obtained porous structures created by LIP, however there is no evidence for it. I recommend BET isotherm or SEM investigation of cross-section to proof this statement. Beside this, it would be nice to calculate the porosity and pore-size diameter, pore-size distribution.
Author Response
Thank you so much for telling me that my thesis is interesting. Thank you so much for telling me that my thesis is interesting.
- Since our samples were first created using LIP, there is no reference (using laser plasma to create colored three-dimensional graphene shapes.
- Resistance in ohm presented here is not suitable for investigation and discussion.
I agree with your opinion. So, I deleted the resistance and surface resistance graph. so I changed it to the conductivity graph as you mentioned. I will explain this in more detail in Question 3. - Conductivity is the reciprocal of resistivity. In order to obtain the specific resistance, the surface resistance must be multiplied by the thickness of graphene. When the speed is 80,100 mm s-1, the thickness is very small. So their resistivity is very large and conductivity is very low.
And why the conductivity is best when laser speed is 40mm/s? plz read red colored senstence
Page 8, 203-211
Figure 7. (e) is a graph of the conductivity of patterns created according to laser speed. As the scanning speed of the laser decreases, the conductivity tends to increase. This again shows that as the scanning speed of the laser decreases, the heat and pressure in the PI increase. It means high fluence produce graphitization more effectively. However, when the laser speed was 20 mm s-1, there was a slightly lower conductivity than when it was 40 mm s-1. It means that when the laser's speed is 20 mm s-1, the energy is so high that the 3-D carbon network is broken and blown away. Therefore, we have confirmed that the speed of 40mm s-1 is the speed of the highest conductivity value. However, this strong energy at laser speed of 20 mm s-1 creates new shapes on the surface. As shown in Figure 7 (f). We observed the unique colorful patterns. We predicted that it is because of structural color of LIPG patterns fabricated by strong heat and pressure.
- I changed all the mm/s that exist in the paper to mm s-1.
- We strongly agree with your opinion. If you look at the cross-section FE-SEM image in Figure 6, you can see that pores are formed. In order to develop the pores, strong heat and pressure are required, which is based on the selective energy concentration of the laser. Polyimide under strong heat and pressure vaporizes and we can confirm LIP phenomenon.
We were satisfied after looking at the images and checking the pores, and we did not measure the size of the pores.
The BET experiment is unlikely to be possible due to the time and circumstances of the thesis review, so please be generous with the results.
Best regards,

Reviewer 2 Report
This manuscript discusses the fabrication of pulsed UV laser-induced porous graphene patterns in polyamide films. By varying the scan speed of the laser beam, interesting structural changes in the film and optical reflectivity were observed. This investigation has also evidenced production of nanospheres in the films. Further, optical and electrical characteristics of the films have been studied. This shows strong originality. However, this manuscript is poorly written (with many English/typo errors) and needs major correction before acceptance. Also, the abstract could be bit more focussed with appropriate technical words. In addition, it might be better if the scan speeds in figures 8(a) and 8(b) are represented by same colours. X & Y - axes for fig. 7(e ) could be included in the figure.
Author Response
Thank you so much for telling me that my thesis is interesting. Thank you so much for telling me that my thesis is interesting.
1. We changed 1page 17line (porous graphitization patternsà laser induced porous graphene (LIPG)).
2. I changed all the mm/s that exist in the paper to mm s-1.
3. I changed figures 8(a) and 8(b) are represented by same colours as u told me.
4. I deleted the resistance and surface resistance graph. so I changed it to the conductivity graph with X&Y axes as you mentioned. so I changed it to the conductivity graph as you mentioned. Conductivity is the reciprocal of resistivity. In order to obtain the specific resistance, the surface resistance must be multiplied by the thickness of graphene. When the speed is 80,100 mm s-1, the thickness is very small. So their resistivity is very large and then conductivity is very low. I explain more (Page 8, 203-211)
Figure 7. (e) is a graph of the conductivity of patterns created according to laser speed. As the scanning speed of the laser decreases, the conductivity tends to increase. This again shows that as the scanning speed of the laser decreases, the heat and pressure in the PI increase. It means high fluence produce graphitization more effectively. However, when the laser speed was 20 mm s-1, there was a slightly lower conductivity than when it was 40 mm s-1. It means that when the laser's speed is 20 mm s-1, the energy is so high that the 3-D carbon network is broken and blown away. Therefore, we have confirmed that the speed of 40mm s-1 is the speed of the highest conductivity value. However, this strong energy at laser speed of 20 mm s-1 creates new shapes on the surface. As shown in Figure 7 (f). We observed the unique colorful patterns. We predicted that it is because of structural color of LIPG patterns fabricated by strong heat and pressure.
Best regards,

Round 2
Reviewer 1 Report
The authors responded adequately and I recommend to accept this manuscript in its revised form.
This manuscript is a resubmission of an earlier submission. The following is a list of the peer review reports and author responses from that submission.
Round 1
Reviewer 1 Report
The manuscript by J.-U. Lee et al. describes a method of producing laser-induced graphene on a flexible polyimide tape. The authors use a defocused leaser source to increase the beam area and show the dependence of laser movement speed on the quality of produced graphene. The material is characterized by XPS, Raman and electrical measurements to assess its quality.
- I am concerned about the novelty of this work. The fabrication of laser-induced graphene on commercially available polyimide tape has been reported many times in literature, including the authors themselves in Ref. 8, where they used the same 355 nm laser source. The novelty of the work is described as using a defocused laser beam and adjusting laser movement speed to produce graphene with optimal quality. However, these parameters are specific to the equipment and may not apply to other equipment in different labs. To address this issue, the authors should instead present a total energy per unit area, or fluence, to provide meaningful information for other researchers. However, the graphene quality dependence on fluence was already extensively studied in ref. 8 and by other groups (J. Tour, Carbon 126 (2018) 472), including XPS, Raman, and electrical measurements. If the authors insist on the novelty of using different parameters, I would suggest submitting this manuscript to an application-focused journal. I do not see the scientific merit of this work to be worth publishing in Nanotechnology.
- The formation of nanospheres at a laser speed of 20 mm/s should be explained. From my experience, the particles shown in SEM image in Figure 8d do not look as graphitic. More information on the structure and composition of nanospheres is required.
- The authors should include calculated fluence for all laser source parameters.
- The explanation of the observed color of the LIG is unsatisfactory. There is not enough of nanoparticles on the surface of graphene to make Rayleigh scattering significant. Following the author's logic, the soot, which also consists of carbon nanoparticles of about the same size, should be blue. Besides, other films produced at higher laser speed also appear bluish in Fig. 6a-d, though no nanospheres are found in corresponding SEM images in Fig. 4. To convince the reviewer, the authors have to collect an optical spectrum of the reflected light.
- English requires some serious work. Statements like "the absorbed energy acts as the surface energy and is transferred to and melted inside the PI by thermal conduction" make no sense, and should be revised.
Author Response
Hello? I'm Junuk Lee
Thank you very much for reviewing my thesis. I was really impressed with your detailed review.
Thanks to your help, the quality of the paper has improved greatly.
- I am concerned about the novelty of this work. The fabrication of laser-induced graphene on commercially available polyimide tape has been reported many times in literature, including the authors themselves in Ref. 8, where they used the same 355 nm laser source. The novelty of the work is described as using a defocused laser beam and adjusting laser movement speed to produce graphene with optimal quality. However, these parameters are specific to the equipment and may not apply to other equipment in different labs. To address this issue, the authors should instead present a total energy per unit area, or fluence, to provide meaningful information for other researchers. However, the graphene quality dependence on fluence was already extensively studied in ref. 8 and by other groups (J. Tour, Carbon 126 (2018) 472), including XPS, Raman, and electrical measurements. If the authors insist on the novelty of using different parameters, I would suggest submitting this manuscript to an application-focused journal. I do not see the scientific merit of this work to be worth publishing in Nanotechnology.
Thanks for the long advice. With this advice, I have pondered the novelty of my thesis. Novelty in my paper is a structural color based on LIPG. Most of the structural colors are made of dielectric material, We have the advantage of being able to make conductive patterns made of LIPG. And it is not yet reported to the academic community that the PI makes the structure color through laser processing.
- The formation of nanospheres at a laser speed of 20 mm/s should be explained. From my experience, the particles shown in SEM image in Figure 8d do not look as graphitic. More information on the structure and composition of nanospheres is required.
We have described in more detail the principles that are generated according to your advice. I would appreciate it if you look at my manuscript.
- The authors should include calculated fluence for all laser source parameters.
I made a table according to your advice and described the spot size and fluence.
- The explanation of the observed color of the LIG is unsatisfactory. There is not enough of nanoparticles on the surface of graphene to make Rayleigh scattering significant. Following the author's logic, the soot, which also consists of carbon nanoparticles of about the same size, should be blue. Besides, other films produced at higher laser speed also appear bluish in Fig. 6a-d, though no nanospheres are found in corresponding SEM images in Fig. 4. To convince the reviewer, the authors have to collect an optical spectrum of the reflected light.
I have rewritten most of the text to keep my novelty as your advice.
Rayleigh scattering doesn't seem to fit, so I deleted it. Therefore, the principle of structural color is explained again with reference papers. I would be grateful if you could refer to my thesis. Also, in pictures 6a-d, I think there is an error too, so I took a picture again and the bluish part disappeared.
Thanks for the advice.
- English requires some serious work. Statements like "the absorbed energy acts as the surface energy and is transferred to and melted inside the PI by thermal conduction" make no sense, and should be revised.
I decided to delete this part because I thought it was strange. Thanks for the advice.
Again,I don't know who you are, but I really appreciate your review.
Best regards, JUNUK LEE

Reviewer 2 Report
The work seems rushed and is maybe publishable under some major revision:
1. Figure 7. XPS analysis is quite poor. The authors should analyse the C1s band to check the sp2/sp3 species
2. Even though the SEM images show graphene like material their Raman spectra do not support this. Did the authors have any explanation for this? From their spectra it seems that the fluorescence is not the reason for this quality.
3. The authors should check their references and improve it
4. In Materials and Methods all the methods (except LIP) are missing
5. In Figure 5 in (b) and (c) the scale bar is missing
6. On caption of figure 4 the (d,i,n) should change to (e,j,o)
Author Response
Hello? I'm Junuk Lee
Thank you very much for reviewing my thesis. I was really impressed with your detailed review.
Thanks to your help, the quality of the paper has improved greatly.
1. Figure 7. XPS analysis is quite poor. The authors should analyse the C1s band to check the sp2/sp3 species
We found and eliminated a big error in xps analysis. And Raman was taken again to analyze the carbon structures.
2. Even though the SEM images show graphene like material their Raman spectra do not support this. Did the authors have any explanation for this? From their spectra it seems that the fluorescence is not the reason for this quality.
We analyzed this structure as LIPG and rewrote it. The reference paper is "Laser-induced porous graphene films from commercial polymers".
As the Raman analysis was re-executed, the fluoresence portion was removed.
3. The authors should check their references and improve it
More than 20 references have been added. Thanks for the advice.
4. In Materials and Methods all the methods (except LIP) are missing
In this section, additional product names and principles have been described in more detail.
5. In Figure 5 in (b) and (c) the scale bar is missing
I attached a scale bar.
6. On caption of figure 4 the (d,i,n) should change to (e,j,o)
Changed as you mentioned. Thank you.
Again,I don't know who you are, but I really appreciate your review.
Best regards, JUNUK LEE

Reviewer 3 Report
- Introduction is too short and motivation, and the possible application of this approach and graphene should be discussed, by matter of how much graphene can be produced using this approach and if the amount is suitable from some application point of view.
- page 2 line 66 abbreviation of LIP is not mentioned, just in abstract and then in introduction no description of this approach (benefits, drawbacks)
- In Fig. 4a,f,k and Fig. 5a the scale bars are not correctly visible, please provide typical scale bars for SEM of FE-SEM measurements. Also description of micrometer is not accurate um is not micro meter, have to be Greek letter.
- Fig. 6e how the resistance measurements correlates with conductivity measurements in Fig. 7a. The Title of Y axis in Fig. 6e need to be added with corresponding units. Units (ohm/sq) is not SI, need to be specified in order to be comparable to other literature.
- Fig. 7a is not mentioned and omitted from discussion in the text. Why 40 mm/s provides best conductivity. Here, SI (S/m) was used it is ok, but it is not compared to the literature, is this approach effective enough, what is the state of the art in the field. In my eyes the conductivity electric or thermal is the most valuable property of graphene.
- Fig. 7c the XPS spectra are not described properly. it is hard to follow what is changing. I can expect the transformation of sp2 and sp3 hybridization similarly as in Raman spectra Fig. 7d, however no discussion, no calculation, even for Raman spectra Id/Ig and in Raman only 40 mm/s shows 2D structure other materials are according to the spectra not 2D.
- page 8, line 206, Table 3 should be Table 4, and it is not mentioned in the text as well as not discussed.
- Authors should decide, if they will use in case of units mm/s or mm s -1, now it is not unified i.e. with Raman
Since the article present lack of scientific discussion, figures are not clear, and not described in the text as well as tables. Characterization is performed (XPS, Raman) but without relevant discussion and I have to state that overall quality is very poor with not mentioned potential application and no discussion of current state of the art. Therefore, I have to reject this article.
Author Response
Hello? I'm Junuk Lee
Thank you very much for reviewing my thesis. I was really impressed with your detailed review.
Thanks to your help, the quality of the paper has improved greatly.
- Introduction is too short and motivation, and the possible application of this approach and graphene should be discussed, by matter of how much graphene can be produced using this approach and if the amount is suitable from some application point of view.
We have modified the introduction in a little more detail as you advised.
- page 2 line 66 abbreviation of LIP is not mentioned, just in abstract and then in introduction no description of this approach (benefits, drawbacks)
Redefined the description of LIP. Thanks for the advice.
- In Fig. 4a,f,k and Fig. 5a the scale bars are not correctly visible, please provide typical scale bars for SEM of FE-SEM measurements. Also description of micrometer is not accurate um is not micro meter, have to be Greek letter.
Corrected. thank u for cheking.
- Fig. 6e how the resistance measurements correlates with conductivity measurements in Fig. 7a. The Title of Y axis in Fig. 6e need to be added with corresponding units. Units (ohm/sq) is not SI, need to be specified in order to be comparable to other literature.
I'm sorry, but I don't understand well, so please explain again.
- Fig. 7a is not mentioned and omitted from discussion in the text. Why 40 mm/s provides best conductivity. Here, SI (S/m) was used it is ok, but it is not compared to the literature, is this approach effective enough, what is the state of the art in the field. In my eyes the conductivity electric or thermal is the most valuable property of graphene.
It is a LIPG (conductive) material with a structural color. Since most of the structural colors are dielectric material, the conductivity is unique.
- Fig. 7c the XPS spectra are not described properly. it is hard to follow what is changing. I can expect the transformation of sp2 and sp3 hybridization similarly as in Raman spectra Fig. 7d, however no discussion, no calculation, even for Raman spectra Id/Ig and in Raman only 40 mm/s shows 2D structure other materials are according to the spectra not 2D.
I found a big error about xps and deleted it completely. Thanks for the advice. Instead, we focused on Raman analysis with a little more focus.
- page 8, line 206, Table 3 should be Table 4, and it is not mentioned in the text as well as not discussed.
i mentioned in my manuscript. thank u
- Authors should decide, if they will use in case of units mm/s or mm s -1, now it is not unified i.e. with Raman
i use this unit units mm/s
Again, I don't know who you are, but I really appreciate your review.
Best regards, JUNUK LEE

Reviewer 4 Report
The manuscript entitled: Novel Flexible Fabrication of 355 nm pulsed UV laser-induced multiple graphene layers with naospheres and their optical and electrical characteristics” concerns the novel approach based on the laser assisted synthesis route of the 2D carbon structure proposed as an alternative to the CVD and chemical exfoliation. The morphology and chemical nature of the obtained material was characterized using SEM and XPS, respectively, while IV-characteristics supports determination of electric properties. Authors described proposed approach as novel but it is not – this kind of approach has been already reported in Small by Chen et al. https://doi.org/10.1002/smtd.201900208 where the utilization of picosecond laser was used to prepare high quality LIG (laser induced graphene). Authors should refer to this work and underline the novelty of their route. However, the whole work is poorly written: there are many repetitions – eg. in section 2.2.; figure 4 caption, conclusions making the whole manuscript hard to read . The referee has impression that the introduction is divided into two parts provided in points 1. and 2.2. Regarding XPS spectra – authors should provide and analyse high resolution XPS spectra not only the survey spectrum since almost no difference is present between them. The analysis of optical properties should be based on the absorbance/reflectance measurements since graphic in fig. 8c, 8d, 8e and photograph in 8f cannot be treated as data providing scientific and the real information concerning the reflection properties depending on the fabrication parameters. Therefore, I cannot support the publication of the submitted manuscript in its present state.
Author Response
Hello? I'm Junuk Lee
Thank you very much for reviewing my thesis. I was really impressed with your detailed review.
Thanks to your help, the quality of the paper has improved greatly.
Authors described proposed approach as novel but it is not – this kind of approach has been already reported in Small by Chen et al. https://doi.org/10.1002/smtd.201900208 where the utilization of picosecond laser was used to prepare high quality LIG (laser induced graphene). Authors should refer to this work and underline the novelty of their route. However, the whole work is poorly written: there are many repetitions – eg.
As you advised, I was able to ponder the novelty about my thesis.
My research is the structure color of LIPG with nanospheres
Most of the papers have been rewritten to give a detailed description of this.
If you check it again, I would really appreciate it.
Regarding XPS spectra – authors should provide and analyse high resolution XPS spectra not only the survey spectrum since almost no difference is present between them.
I found a big error about XPS and removed it. Raman analysis, a carbon-based material analysis tool, was retaken to confirm graphene's structure .
The analysis of optical properties should be based on the absorbance/reflectance measurements since graphic in fig. 8c, 8d, 8e and photograph in 8f cannot be treated as data providing scientific and the real information concerning the reflection properties depending on the fabrication parameters.
Most of the photos you mentioned were deleted. In addition, various patterns with different structural colors were shown, and the principle was explained with reference.
Again,I don't know who you are, but I really appreciate your review.
Best regards, JUNUK LEE

Reviewer 5 Report
Dear Authors
The paper is reporting some interesting results about Laser Induced Porous Graphene, that is laser induced graphitization of polymers.
The paper requires major amendments before considering the publication on nanomaterials. The main novelty is the observation of light scattering from the porous and nanostructured films after UV irradiation.
There are some errors and the English language is not sufficiently precise, generic words are often used instead of technically appropriate terms, the optical interpretation and analysis are sloppy and should be improved before considering publication.
For names and definitions I recommend to refer to “All in the graphene family – A recommended nomenclature for two-dimensional carbon materials”. In the literature this material is called Laser Induced porous Graphene and not Laser Induced Graphene and the work “Laser-induced porous graphene films from commercial polymers” NATURE COMMUNICATIONS | 5:5714 should be added to the references
I believe that the process should be rather called laser induced graphitization unless it is clearly demonstrated by Raman the production of thin crystalline graphene layers. However due to the porous and structured geometry of their films and to the poor Raman signature, authors should at least call their process and material Laser Induced porous Graphene .
Although the authors propose the application to flexible electronics, no real application is demonstrated nor proposed. Moreover neither flection nor adhesion tests were performed.
Abstract
The abstract is vague and generic. It should be improved to clearly resume the paper content
Page 1 line 17 not clear the meaning of unique, please be more specific
Page 1 line 17-18 Text is not very clear. In the abstract it should be clearly stated that nanospheres and porous graphene structures were created during LIG process
Introduction
Page 1 line 38, about LIG, there is no Raman evidence for graphene in the paper, either Laser Induced porous Graphene or Laser Induced Graphitization.
Page 2 line 46 A reference for the absorption of a single layer should be added. Few layer graphene is partially transmitting also.
Page 2 line 48-55 This whole section is a confusing, uses unscientific terms (bright, dazzling, high,…) and should be rewritten. The spectral transmission of graphene is mostly flat, so graphene's colors mentioned here are not true colors.
Page 2 line 50 See previous note. Substrates with transparent oxides topped with GO and G cause light interference, which depend on wavelength, giving apparent colour to reflected white light. Investigation often refer to metal-oxide-graphene structures such as Si/SiO2/graphene
Page 2 line 52 Have and not has
Page 2 line 54 black silicon is black due to multiple reflections, independently on the presence of graphene. Spectral reflectance depends on the details of the substrate layer structure and on the refractive indices.
Page 2 line 56 Colors in this case will depend on the photonic structures (graphene flakes and nanoshperes) created during LIG. “Structural colours” does not make sense, are they due to absorbance, reflectance, scattering?
Page 2 line 64 “We demonstrated highly flexible and conductive LIG patterns for wearable opto-electronics.” Only electrical conductance is demonstrated, no application is proposed in the paper.
Materials and Methods
Page 2 line 66 and 71 define LIP and LDW acronyms both in text and abstract
Page 3 line 85 Imide group. not ring
Page 3 line 87-89 The sentence is unclear and it should be reworded
Page 3 line 96 “absorption rate” is not an appropriate physical quantity. Absorbance depends on Absorption coefficient and thickness, please apply correct physical terms and quantities
Page 3 line 98 cleaner with respect to what? The “clean” meaning is unclear and it appears colloquial.
Page 4 line 105 break and not remove
Page 4 line 107 Continues? Does is mean “increases”
Page 4 line 108 LIP is plasma ablation, photochemical ablation is due to laser photons detaching atoms or group directly
Page 4 lines 114-116 LIP threshold depends on power and absorption. Authors should explain why it depends on overlap. Is this due to the increase of absorption of laser light due to progressive graphitization?
Page 5 line 118 Please describe laser focusing apparatus (lens type, diameter and focal length)
Page 5 line 124 LIP threshold depends on the intensity of each single pulse and on the sample surface.
Results and discussion
Page 5 line 134-136 Very unclear, please reword applying physical values and without using colloquial words such as “certain”
Page 5 line 136-146 This section should be improved. At lower speed overlapping increases, there is more overall deposited energy, the light intensity does not change, but the sample surface is progressively modified.
Page 6 line 162-163 There is no evidence for this effect in the paper, add a reference
Page 6 line 163 There is no evidence of a graphene layer, graphene is cristalline and one atom thick, this is rather a graphitization process. Porous graphene is produced according to SEM.
Page 7 figure 6 Please add markers to the micrographs
Page 7 line 173 Colours due to scattering
Page 7 line 180 The laser pulse overlap increased
Page 7 line 183 “was removed a little” in what sense, is there evidence for etching?
Page 7 lines 184-185 Deduced and not predicted. This sentence should be moved below, after figure 8.
Page 8 figure 7a The graph should contain also the values for 100 and 80mm/s scanning speeds
Page 8 Figure 7d and lines 202 203 These Raman spectra show either amorphous carbon or very highly defective carbonaceous materials. These Raman spectra are much worse than on the reference work “Laser-induced porous graphene films from commercial polymers”. This should be explicitly mentioned in the text, the process is a laser induced graphitization and graphene is far away. For instance, the Raman bands are much wider than in the paper “Laser-induced porous graphene films from commercial polymers”.
Page 9 figure 8 Please omit figure 8c
Also Figure 8d The reflection and refraction scheme is badly wrong, since it is good for objects much bigger than the wavelength, and should be also omitted. Is the particles are below 1/10 the wavelength this scheme is not correct and a scattering theory (such as Rayleigh or Mie theories) should be applied
Also figure 8e is difficult to understand. Where are refraction and refraction occurring? If this is the model, then colour should change with angle and show a rainbow effect. If it is a Rayleigh back scattering (bottom of the figure) the scheme on the top is difficult to understand. Add a simple scheme showing that in the case of Rayleigh scattering the blue back-reflection is enhanced.
Page 9 lines 215-221 The meaning of the word amplification is unclear. Figure 8d is wrong and should be omitted
Page 9 lines 223 Please improve the discussion of Rayleigh scattering. “I” is the intensity of the scattered light.
Page 9 line 230-232 Reword the sentence. Is the conclusion that scattering is from nanoshperes? Or from the porous graphitic structure?
Conclusions
should be modified according to the paper improvements
Author Response
Hello? I'm Junuk Lee
Thank you very much for reviewing my thesis. I was really impressed with your detailed review.
Thanks to your help, the quality of the paper has improved greatly.
With the first mention of "Laser-induced porous graphene films from commercial polymers," I decided to change from title to LIG to LIPG.
Abstract
As you mentioned, I wrote it in more detail.
Introduction: I excluded most of the ambiguous content.
Page 1 line 38, about LIG, there is no Raman evidence for graphene in the paper, either Laser Induced porous Graphene or Laser Induced Graphitization. -> I wrote down (Raman evidence).
Page 2 line 46 A reference for the absorption of a single layer should be added. Few layer graphene is partially transmitting also. -> I changed as you mentioned. Thank u.
Page 2 line 48-55 This whole section is a confusing, uses unscientific terms (bright, dazzling, high,…) and should be rewritten. The spectral transmission of graphene is mostly flat, so graphene's colors mentioned here are not true colors. -> I write down technical form instead of unscientific term as u told me. u can check introduction of my manuscript.
Page 2 line 56 Colors in this case will depend on the photonic structures (graphene flakes and nanoshperes) created during LIG. “Structural colours” does not make sense, are they due to absorbance, reflectance, scattering? Our conclusion is that structural color is because of multiple scattering causing interference based on LIPG : unique structure (multiple layer and nanopsheres).
Page 2 line 64 -> I deleted this sentence
Materials and Methods
Page 2 line 66 and 71 define LIP and LDW acronyms both in text and abstract-> I defined both of them.
Page 3 line 85 Imide group. not ring -> I deleted this word.
Page 3 line 87-89 The sentence is unclear and it should be reworded
Page 3 line 96 “absorption rate” is not an appropriate physical quantity. Absorbance depends on Absorption coefficient and thickness, please apply correct physical terms and quantities-> I write down with reference. Can u check again?
Page 3 line 98 cleaner with respect to what? The “clean” meaning is unclear and it appears colloquial. – > deleted.
Page 4 line 105 break and not remove -> I deleted
Page 4 line 107 Continues? Does is mean “increases” : yes it is I changed to “increase”
Page 4 line 108 LIP is plasma ablation, photochemical ablation is due to laser photons detaching atoms or group directly. -> yes it is . so I changed as u told me.
Page 4 lines 114-116 LIP threshold depends on power and absorption. Authors should explain why it depends on overlap. Is this due to the increase of absorption of laser light due to progressive graphitization? Yes it is .
Page 5 line 118 Please describe laser focusing apparatus (lens type, diameter and focal length)
->I wrote down with product name and focal length.
Page 5 line 124 LIP threshold depends on the intensity of each single pulse and on the sample surface. ->Yes but I don’t know the exact threshold value. Thank u for check.
Results and discussion
Most of the results were deleted and rewritten.
Therefore, it would be good to see it from the beginning again.
Conclusion
in the same way of results and discussion.
Again,I don't know who you are, but I really appreciate your review.
Best regards, JUNUK LEE

Round 2
Reviewer 3 Report
The manuscript was revised by authors, unfortunately still, does not have the quality for publication in the Nanomaterials MDPI. Please see the comments below.
- Original question was not responded adequately, the LIP approach for graphene application is still not clear, any suggestions and potential application have not been mentioned.
- This question was revised accordingly.
- This remark was revised accordingly.
- Resistance in ohm presented here is not suitable for investigation and discussion. Conductivity measurements need to be performed to be able to provide comparative method to Raman spectroscopy. Unit of Surface Resistance is not comparable to majority of the articles, how the unit ohm/sq is defined? What is sq? 1 m2 or 1 km2. Moreover, difference in resistance is just two orders of magnitude, is it reduction that was obtained similar or comparable also for the rest of the laser-induced approaches? References are missing here.
- Response to original question no. 5 was not adequate. The question was, what physical change is behind the result that 40 mm/s has the best conductivity
- Delete of the spectra is not scientifically correct. Your manuscript have to be carefully prepared from the beginning and if something have to be published or not published need to be adequately discussed. Response in the way, "I saw error and had deleted it" is not correct.
- This question was revised accordingly.
- OK, then the Raman spectra need to be on X axis changed, cm-1 is not possible ones you decided to mm/s.
- Moreover, the authors claimed some porous structures created by LIP, however there is no evidence for it. Some BET or SEM investigation of cross-section of the material? Or if there is some porosity, it would be nice to calculate the porosity and pore-size diameter, some distribution curve.
Reviewer 4 Report
Still I keep my decision that manuscript is poorly written and needs substantial improvement of style and grammar the same as received answers from the corresponding authors. The structure of many sentences is very strange, eg the last sentence of the abstract: And, we applied these unique characteristics to various colorful patterns by controlling the focal plane. The sentence on page 5 starts with a small letter and after the comma word "we" is given with capital letters and in many places various fonts are used or numbers are not provided correctly in superscripts. Authors wrote also that "patterns exhibit stable electrical behaviour" - what authors have on mind writing 'pattern' - it is the whole material or only part? There are many of such controversial and unclear issues needing further improvement. The whole article is full of such mess that should not appear in scientific article.
Moreover, from the chemical point of view structures provided in part c) do not exist, eg. C2, CN as it is provided in Fig. 2 - it is in radical/ion form? In the part b) of fig. 2 - the given H symbol is also unclear-what is its nature: is it radical/ion? In my opinion, the material is poorly studied, since only SEM and Raman analysis is provided.
Summarizing, I cannot support the publication of this paper in Nanomaterials issue.
Reviewer 5 Report
Dear Authors,
The paper has improved. However still many changes should be made, the English language should be improved and Raman analisys also:
Page 2 line 44-45 “The delamination effect (photochemical effect) occurs when the UV laser power is weak, while the carbonization effect (photothermal effect) occurs as the laser fluence increases.“ Please reword and add a reference
Page 2 line 55 “the commonly used single or few layer graphene is transparent due to the Pauli barrier effect [15, 16].” All thin materials are partially transmitting due to incomplete absorption of light
Page 2 line 57 change “structure color” in "structural coloration" over the whole paper
Page 2 line 63 “and it has been mainly studied at tens of nm thickness”. It has been studied for dielectric thickness of several hundreds of nm
Page 2 line 63 “The reflectance varies according to the thickness or the dielectric layer, but the reflectance in the blue region near 400 nm is high” The colouring depends on the thickness
Page 2 line 67 “reflectance of the graphene multilayers” furthe in the paper it is written that structural colouring is due to Rayleigh scattering from nanospheres
Page 2 line 75-76: Not well written and unclear, delete or reword
Page 4 line 118-122: improve the English language
Page 4 line 127 “Since we used LDW by carrying a laser beam” badly written “performed the LDW process by moving the laser beam”
Page 5 line 137 delete “rate”
Page5 line 137 explain why “We expect that this high pulse overlap rate made the pattern’s structure unique, which is different from the findings in other studies”
Page 5 line 142-150 Improve the language
Page 6 line 166 Unclear when it was predicted?
Page 8 line 210 colourful patterns, structural colouration
Page 8 figure 8. Graphene quality apparently increases with scanning speed, since the bands are becoming narrower. At low scanning speed the material appera mostly amorphous.
Page 8 line 214 "demonstrates” shows
Page 8 line 226-227 “Therefore, the ??/?? values (Table 5) were also used to specify the graphitization level.” Id/Ig measures the defectiveness of carbon films. Graphitization is measured by FWHM of the G line. The Raman analysis should be considerably improved.
Page 9 line 240 structural coloration